# Structure of the two-component S-layer of the archaeon *Sulfolobus acidocaldarius*

Lavinia Gambelli[1,2†], Mathew McLaren[1,3], Rebecca Conners[1,3], Kelly Sanders[1,3], Matthew C Gaines[1,3], Lewis Clark[1,3‡], Vicki AM Gold[1,3], Daniel Kattnig[1,2], Mateusz Sikora[4,5], Cyril Hanus[6,7], Michail N Isupov[8], Bertram Daum[1,3]*

[1]Living Systems Institute, University of Exeter, Exeter, United Kingdom; [2]Faculty of Environment, Science and Economy, University of Exeter, Exeter, United Kingdom; [3]Faculty of Health and Life Sciences, University of Exeter, Exeter, United Kingdom; [4]Department of Theoretical Biophysics, Max Planck Institute for Biophysics, Frankfurt, Germany; [5]Malopolska Centre of Biotechnology, Jagiellonian University, Kraków, Poland; [6]Institute of Psychiatry and Neurosciences of Paris, Inserm UMR1266 - Université Paris Cité, Paris, France; [7]GHU Psychiatrie et Neurosciences de Paris, Paris, France; [8]Henry Wellcome Building for Biocatalysis, Biosciences, Faculty of Health and Life Sciences, University of Exeter, Exeter, United Kingdom

**Abstract** Surface layers (S-layers) are resilient two-dimensional protein lattices that encapsulate many bacteria and most archaea. In archaea, S-layers usually form the only structural component of the cell wall and thus act as the final frontier between the cell and its environment. Therefore, S-layers are crucial for supporting microbial life. Notwithstanding their importance, little is known about archaeal S-layers at the atomic level. Here, we combined single-particle cryo electron microscopy, cryo electron tomography, and Alphafold2 predictions to generate an atomic model of the two-component S-layer of *Sulfolobus acidocaldarius*. The outer component of this S-layer (SlaA) is a flexible, highly glycosylated, and stable protein. Together with the inner and membrane-bound component (SlaB), they assemble into a porous and interwoven lattice. We hypothesise that jackknife-like conformational changes in SlaA play important roles in S-layer assembly.

**\*For correspondence:**
b.daum2@exeter.ac.uk

**Present address:** †Medical Research Council Laboratory of Molecular Biology, Cambridge, United Kingdom; ‡Department of Cellular Biochemistry, University Medical Center Göttingen, Göttingen, Germany

**Competing interest:** The authors declare that no competing interests exist.

## Editor's evaluation

The manuscript brings important new insights in S-layer structure and assembly, providing a first experimental model of a crenarchaeotal S-layer. The work provides solid evidence for the S-layer architecture and its role in supporting the archaeal cell envelope. This will be of broad interest to microbiologists and biotechnologists seeking to understand the biological role and technological application of these enigmatic membrane support structures.

## Introduction

The prokaryotic cell envelope includes a cytoplasmic membrane and a cell wall, which provide structural integrity to the cell and mediate the interaction between the extracellular and intracellular environment. The cell wall differs in composition and structure across prokaryotes (*Bharat et al., 2021*). In bacteria, a peptidoglycan (murein) layer encapsulates the cytoplasmic membrane, and this is in turn enclosed by a second membrane in Gram-negative bacteria (*Fagan and Fairweather, 2014*). Generally, the archaeal cell wall lacks an outer membrane, but a variety of cell wall elements, including pseudomurein, methanochondroitin, and protein sheaths have been described (*Klingl et al., 2019*). Most prokaryotes exhibit a porous glycoprotein surface layer (S-layer) as the outermost component of

their cell wall (*Bharat et al., 2021*). In archaea, S-layers are the simplest and most commonly found cell wall structure (*Bharat et al., 2021*; *Klingl et al., 2019*; *Albers and Meyer, 2011*; *Rodrigues-Oliveira et al., 2017*).

The prokaryotic cell envelope is exposed to a variety of environmental conditions, which, in the case of extremophiles, can be unforgiving (low/high pH, high temperature, and salinity). Therefore, S-layers reflect the cellular need for both structural and functional plasticity, allowing archaea to thrive in diverse ecosystems. Archaeal S-layers maintain the cell shape under mechanical, osmotic, and thermal stress, selectively allow molecules to enter or leave the cell, and create a quasiperiplasmic compartment (similar to the periplasmic space in Gram-negative bacteria) (*Klingl et al., 2019*; *Albers and Meyer, 2011*; *Rodrigues-Oliveira et al., 2017*). S-layer glycoproteins are also involved in cell–cell recognition (*Shalev et al., 2017*) and mediate virus–host interactions (*Tittes et al., 2021*; *Schwarzer et al., 2023*).

Structurally, an S-layer is a pseudocrystalline array of (glyco)proteins (surface layer proteins, SLPs). The ordered nature of an S-layer is what sets it apart from other protein sheaths (*Bharat et al., 2021*; *Fagan and Fairweather, 2014*; *Klingl et al., 2019*; *Sleytr et al., 2014*). S-layers usually consist of thousands of copies of one SLP species. These SLPs self-assemble on the cell surface predominantly at mid-cell (*Bharat et al., 2021*; *Abdul-Halim et al., 2020*), giving rise to an oblique (p1, p2), square (p4), or hexagonal (p3, p6) symmetry (*Sleytr et al., 2014*). In archaea, the hexagonal symmetry is the most common (*Albers and Meyer, 2011*). The S-layer is highly porous. Depending on the species, the pores can occupy up to about 70% of the S-layer surface and have different sizes and shapes (*Albers and Meyer, 2011*; *Sleytr et al., 2014*). Such an assembly provides a highly stable and flexible 2D lattice (*Engelhardt and Peters, 1998*; *Engelhardt, 2007*). Archaeal SLPs range from 40 to 200 kDa in molecular mass and show little sequence conservation (*Bharat et al., 2021*). The most common post-translational modification of SLPs is glycosylation. Most archaeal SLPs are *N*- and/or *O*-glycosylated and the composition of the glycans is highly diverse (*Albers and Meyer, 2011*; *Rodrigues-Oliveira et al., 2017*). Thermophilic and hyperthermophilic archaea show a higher number of glycosylation sites on SLPs compared to mesophilic archaea, suggesting that glycans support thermostability (*Meyer and Albers, 2013*). Another common aspect of archaeal S-layers is their binding of divalent metal ions (*Engelhardt, 2007*; *Cohen et al., 1991*; *von Kügelgen et al., 2021*), which have been shown to be essential for S-layer assembly and anchoring in bacteria (*Herdman et al., 2022*; *Baranova et al., 2012*). Atomic models of assembled bacterial S-layers have been reported, including that of *Clostridium difficile* (*Lanzoni-Mangutchi et al., 2022*), *Caulobacter crescentus* (*Bharat et al., 2017*; *von Kügelgen et al., 2020*), and *Deinococcus radiodurans* (*von Kügelgen et al., 2023*), However, archaeal S-layers have been less well explored at this level of detail. So far, atomic models for domains of *Methanosarcina* SLPs (*Jing et al., 2002*; *Arbing et al., 2012*), and more recently, a structure of the Euryarchaeon *Haloferax volcanii* S-layer have been described (*von Kügelgen et al., 2021*).

*Sulfolobus acidocaldarius* is a hyperthermophilic and acidophilic Crenarchaeon and thrives in acidic thermal soils and hot springs worldwide. It grows at pH ~2–3 and temperatures ranging from 65 to 90°C (*Brock et al., 1972*). The *Sulfolobus* S-layer is composed of two repeating glycoproteins, SlaA and SlaB. In *S. acidocaldarius*, SlaA contains 1424 amino acids and has a molecular mass of 151 kDa, whereas SlaB comprises 475 amino acids and has a mass of 49.5 kDa (*Grogan, 1996*). Comparative sequence analysis and molecular modelling predicted that SlaA is a soluble protein rich in β-strands (*Veith et al., 2009*). On the other hand, SlaB has been predicted to contain three consecutive β-sandwich domains at the N-terminus and a membrane-bound coiled-coil domain at the C-terminus (*Veith et al., 2009*). Across the Sulfolobales, SlaA shows higher sequence and structural variability compared to SlaB (*Veith et al., 2009*). Early 2D crystallography and electron microscopy experiments described the *S. acidocaldarius* S-layer as a 'smooth', highly porous, hexagonal (p3) lattice (*Grogan, 1996*; *Taylor et al., 1982*). Recently, we investigated the architecture of the *S. acidocaldarius* S-layer by cryo electron tomography (cryoET) (*Gambelli et al., 2019*). The S-layer has a bipartite organisation with SlaA and SlaB forming the extracellular- and intracellular-facing layers, respectively. Dimers of SlaA and trimers of SlaB assemble around hexagonal and triangular pores, creating a ~30-nm-thick canopy-like framework. However, the resolution was limited, and secondary structure details were unresolved. *Sulfolobus* mutants lacking SlaA and/or SlaB show morphological aberrations, higher sensitivity to hyperosmotic stress and alterations of the chromosome copy number, suggesting that in these species the S-layer plays key roles in cell integrity, maintenance, and cell division (*Zhang et al., 2019*).

Here, we studied the *S. acidocaldarius* S-layer and its components using a combination of single-particle cryo electron microscopy (cryoEM) and cryoET. We solved the atomic structure of SlaA and investigated its stability across extreme pH ranges. Moreover, we combined cryoEM data and Alphafold2 to build a complete in situ atomic model of this S-layer and propose insights into its dynamics and assembly.

## Results

### Structure and *N*-glycosylation of SlaA$_{30-1069}$ at acidic pH

To solve the structure of the *S. acidocaldarius* SLP SlaA, we disassembled the S-layer by changing the pH from acidic to basic and purified the native protein using size-exclusion chromatography. We have previously shown that *S. acidocaldarius* SlaA purified in this way reforms S-layers upon shifting the pH back to acidic (*Gambelli et al., 2019*). This demonstrates that after disassembly, SlaA remains in a 'native', reassembly competent form.

CryoEM grids with suspensions of the protein were plunge frozen at pH 4, before the protein had time to reassemble into S-layers. The acidic pH was chosen to account for the natural conditions in which *S. acidocaldarius* thrives. The structure of SlaA was determined from cryoEM movies, using the single-particle analysis (SPA) pipeline in Relion 3.1 (*Scheres, 2020*; *Figure 1—figure supplement 1*, *Figure 1—figure supplement 2a, d*; *Supplementary file 1a, d* ). The final cryoEM map had a global resolution of 3.1 Å (*Figure 1—figure supplement 3a, b* and *Figure 1—figure supplement 4a*).

Because SlaA has virtually no homology with other structurally characterised proteins, the cryoEM map was used to build an atomic model de novo (*Figure 1a*; *Figure 1—figure supplement 4b*; *Video 1*). Residues 30–1069 (~70% of the sequence) were clearly defined in the cryoEM map. The N-terminal signal peptide (predicted to be residues 1–24) is cleaved prior to S-layer assembly (*Veith et al., 2009*). A few N-terminal residues and residues 1070–1424 at the C-terminus were not resolved by SPA, likely due to their high flexibility (*Figure 1—figure supplement 5a*; *Video 2*). SlaA$_{30-1069}$ is a Y-shaped protein. It consists mostly of β-strands and contains only a few short α-helices (*Figure 1a, b*, *Figure 1—figure supplement 4c, d*). The polypeptide chain is arranged into four domains (D1$_{30-234}$, D2$_{235-660,701-746}$, D3$_{661-700,747-914}$, and D4$_{915-1069}$), as defined by SWORD (*Postic et al., 2017*; *Figure 1c*).

Of those domains, only D4 shows significant similarity to known structures – the domain 3 of complement C5 (PDB ID: 4E0S) according to DALI (*Holm, 2020*). A disulphide bond links D3 and D4 (Cys$_{677}$–Cys$_{1017}$) (*Figure 1—figure supplement 4d*), however, the density of this bond is not visible in the cryoEM map, likely due to electron beam damage (*Kato et al., 2021*).

The structure of the missing C-terminus (SlaA$_{914-1424}$) was predicted (including D4 to aid alignment) using Alphafold (*Jumper et al., 2021*) and revealed two additional β-domains, D5 and D6 (*Figure 1c*, *Figure 1—figure supplement 6*). Alphafold predicted five different conformations of SlaA$_{914-1424}$, which differed with regard to the position of D5–D6 relative to D1–D4, suggesting an in-plane flexibility between these two parts of the protein around a hinge (amino acids A$_{1067}$–L$_{1071}$) between D4 and D5 (*Figure 1c*, *Figure 1—figure supplement 6*). Similar conformations were also observed in 2D classes of our cryoEM dataset (*Figure 1—figure supplement 5a*, *Video 2*), as well as a low-resolution 3D refinement of SlaA purified from the related species *Saccharolobus solfataricus* (*Figure 1—figure supplement 5b, c*), substantiating the Alphafold predictions in *Figure 1—figure supplement 6*. The predicted extremes of the conformational space of SlaA are shown in *Figure 1c, d*. These describe stretched (open) and flapped (closed) conformations. The highly variable positions of D5–D6 seen in the 2D classes, suggest that these domains do not adopt discrete positions, but rather move about freely in the soluble form of the SlaA subunit. It is probable that this jackknife-like flexibility aids SlaA's assembly into an interwoven S-layer. If some of this flexibility is retained in the assembled S-layer, it will enable it to adopt various degrees of curvature, necessitated by its ability to encapsulate large cells, as well as small exosomes.

SlaA is expected to be highly glycosylated; its sequence contains 31 predicted *N*-glycosylation sites (*Peyfoon et al., 2010*). Our cryoEM map of SlaA$_{30-1069}$ shows 19 glycan densities (*Figure 2*), largely in agreement with the prediction of 20 sequons located in this portion of the protein (*Peyfoon et al., 2010*). The 19 glycosylated Asn residues in SlaA$_{30-1069}$ are listed in *Figure 2e*. The remaining predicted glycosylation sites reside in domains D5 and D6, in which eight sites were confirmed to be

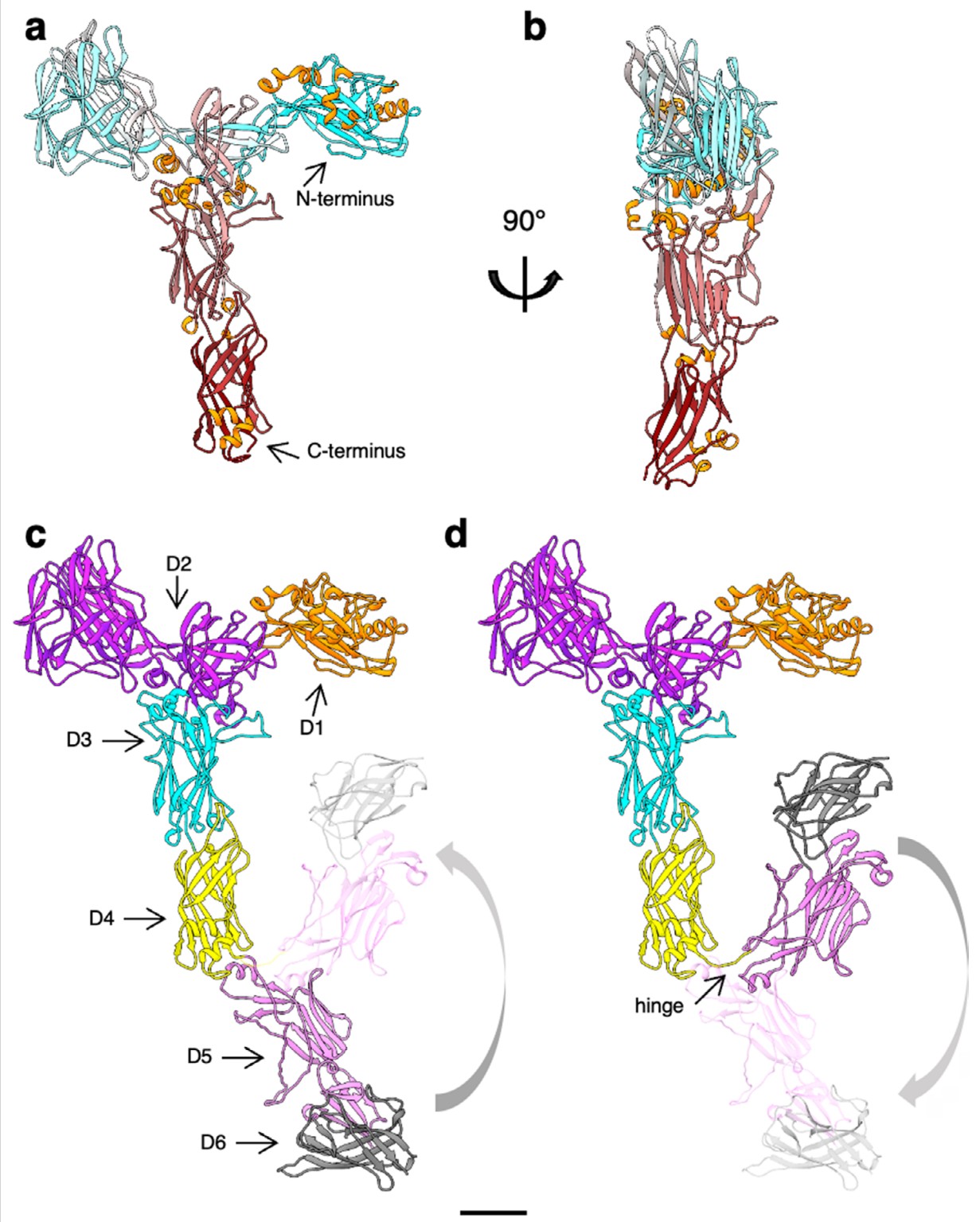

**Figure 1.** Atomic model of *S. acidocaldarius* S-layer protein SlaA at pH 4. (**a, b**), SlaA$_{30-1069}$ atomic model obtained by single-particle cryo electron microscopy (cryoEM) in ribbon representation and cyan–grey–maroon colours (N-terminus, cyan; C-terminus, maroon) with α-helices highlighted in orange. (**c, d**) SlaA atomic models highlighting six domains: D1$_{30-234}$ (orange), D2$_{235-660,701-746}$ (purple), D3$_{661-700,747-914}$ (cyan), D4$_{915-1074}$ (yellow), D5$_{1075-1273}$ (pink), and D6$_{1274-1424}$ (grey). D5 and D6 were predicted using Alphafold. A flexible hinge exists between D4 and D5. D5 and D6 are thus free to move relative to D1–D4 in the isolated SlaA particle (represented by a curved grey arrow between a stretched (**c**) and a flapped (**d**) conformation). Scale bar, 20 Å.

*Figure 1 continued on next page*

*Figure 1 continued*

The online version of this article includes the following figure supplement(s) for figure 1:

**Figure supplement 1.** Relion processing workflow for the pH 4 dataset of SlaA.

**Figure supplement 2.** Representative cryoEM micrographs and 2D classes for SlaA.

**Figure supplement 3.** Resolution estimation for the cryoEM maps of SlaA.

**Figure supplement 4.** SlaA$_{30-1069}$ cryo electron microscopy (cryoEM) map and atomic model.

**Figure supplement 5.** SlaA flexibility.

**Figure supplement 6.** Five Alphafold predictions of SlaA$_{914-1424}$.

glycosylated by mass spectrometry analysis (*Peyfoon et al., 2010*). Therefore, the entire SlaA protein contains a total of 27 confirmed glycans.

The *N*-glycans were modelled into the cryoEM densities based on their known chemical structure (*Zähringer et al., 2000*). The complete glycan is a tribranched hexasaccharide, containing a 6-sulfoquinovose (QuiS). Not all glycosylation sites had clear density to model the entire hexasaccharide. Instead, several forms of apparently truncated glycans were fitted into the cryoEM map (*Figure 2b–d*). Most glycans (47 %) were built as pentasaccharides, lacking the glucose bound to QuiS in the mature glycan; 15% of the glycan pool could be modelled with the whole hexasaccharide structure.

As shown for other glycoproteins, such as the spike proteins of coronavirus (*Sikora et al., 2021*), glycans are usually much more dynamic than polypeptides and rapidly explore large conformational spaces, generating potentially bulky glycan shields over hundreds of nanoseconds. To evaluate the morphology and span of such shields, a reductionist molecular dynamics simulation approach (GlycoSHIELD) (*Gecht et al., 2021*) was used to graft plausible arrays of glycan conformers onto open and closed conformations of SlaA monomers with D5 and D6 domains (*Figure 2g, h*). Glycan volume occupancy was comparable on the two conformations of the monomers (*Figure 2g, h*).

Both closed and open conformations showed a similar number of possible glycan conformers (with the closed slightly more than the open form; *Figure 2—figure supplement 1*). This signifies that neither SlaA conformation is entropically favoured over the other, which allows for the observed free jackknife movement between D1–4 and D5–6 (*Video 2*).

## SlaA at different pH conditions

SlaA assembly and disassembly are pH-sensitive processes (*Gambelli et al., 2019*). A pH shift from acidic (~pH 4) to alkaline (~pH 10) induces the

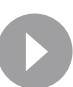

**Video 1.** Atomic structure and glycosylation of SlaA$_{30-1069}$. The SlaA$_{30-1069}$ cryo electron microscopy (cryoEM) map is shown in cornflower blue. The atomic structure is shown in ribbon representation and coloured in cyan–grey–maroon. N-terminus, cyan; C-terminus, maroon. The glycosylated Asn residues are in orange and the glycans are represented as balls and sticks. C, medium blue; N, dark blue; O, red; S, yellow.

https://elifesciences.org/articles/84617/figures#video1

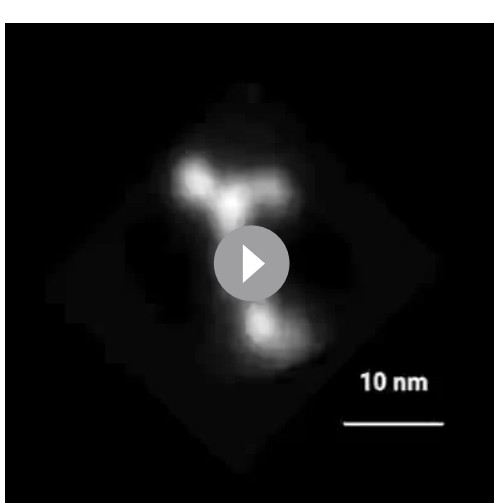

**Video 2.** Flexibility of SlaA. Sequence of 2D classifications of negatively stained SlaA obtained in Relion 3. D2-4 were aligned, showing the flexibility of D1, D5, and D6.

https://elifesciences.org/articles/84617/figures#video2

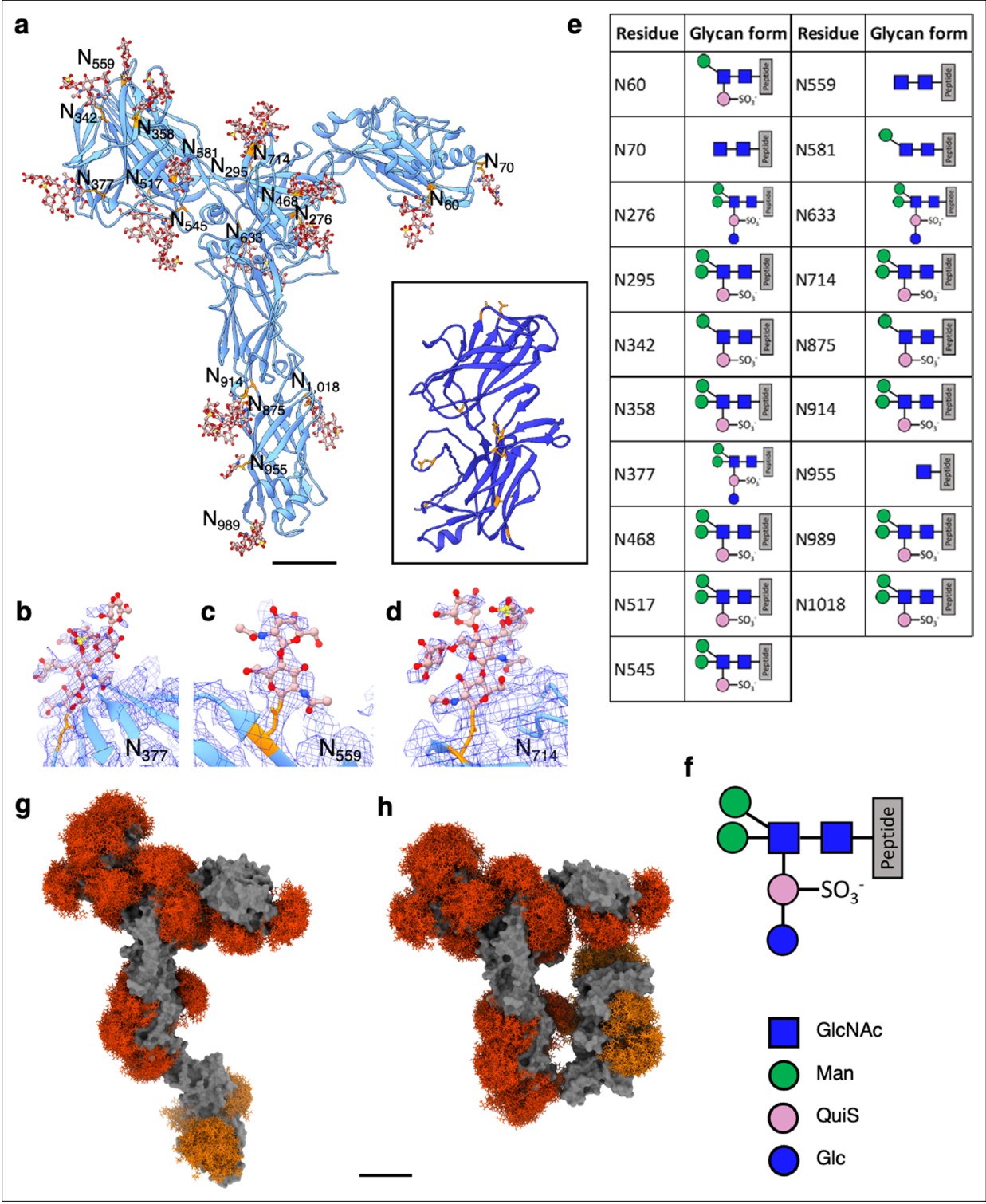

**Figure 2.** *N*-glycosylation of *S.acidocaldarius* SlaA. (**a**) Atomic model of SlaA in ribbon representation. SlaA$_{30–1069}$ as solved by cryoEM is in cornflower blue; SlaA$_{1070–1424}$ as predicted by Alphafold is in purple (boxed). 19 Asn-bound N-glycans were modelled into the cryoEM map of in SlaA$_{30–1069}$ (glycans rusty brown sticks, Asn in orange). In the glycans, O atoms are shown in red, N in blue, and S in yellow. The inset shows the Alphafold model of SlaA$_{1070–1424}$ (D5 and D6), where eight likely glycosylated Asn residues (*Peyfoon et al., 2010*) are highlighted as orange sticks. Scale bar, 20 Å. (**b–d**) Example close-ups of glycosylation sites with superimposed cryoEM map (blue mesh). (**b**) Shows the full hexasaccharide on Asn$_{377}$, (**c**) shows GlcNAc$_2$ on Asn$_{559}$, and (**d**) shows a pentasaccharide lacking Glc$_1$ on Asn$_{714}$. (**e**) List of glycosylation sites and associated glycans of SlaA$_{30–1069}$. The schematic glycan representation (**f**) is equivalent to (*Peyfoon et al., 2010*). Blue square, *N*-acetylglucosamine; green circle, mannose; pink circle, 6-sulfoquinovose; blue

*Figure 2 continued on next page*

*Figure 2 continued*

circle, glucose. (**g, h**) GlycoSHIELD models (red, orange) showing the glycan coverage of the protein (solid grey). Glycan shields corresponding to glycosylation sites visualised by cryoEM are coloured red, glycan shields with the Alphafold model of the SlaA C-terminus are shown in orange.

The online version of this article includes the following figure supplement(s) for figure 2:

**Figure supplement 1.** Entropic contribution of glycans to protein conformation.

disassembly of the lattice into its component subunits, while a reassembly occurs upon shifting the pH back to acidic (*Gambelli et al., 2019*). Asking whether this pH shift-induced assembly and disassembly mechanism is based on a conformational change or partial unfolding of SlaA, we investigated the structure of SlaA at different pH conditions. Purified SlaA proteins were frozen at pH 7 and 10 and their structure was determined using the SPA pipeline in Relion (*Zivanov et al., 2018*; *Figure 3—figure supplement 1a, b*; *Supplementary file 1a*) and 3.1 (*Figure 3—figure supplement 2*, *Supplementary file 1a*; *Figure 3—figure supplements 1 and 2*). The resulting cryoEM maps had global resolutions

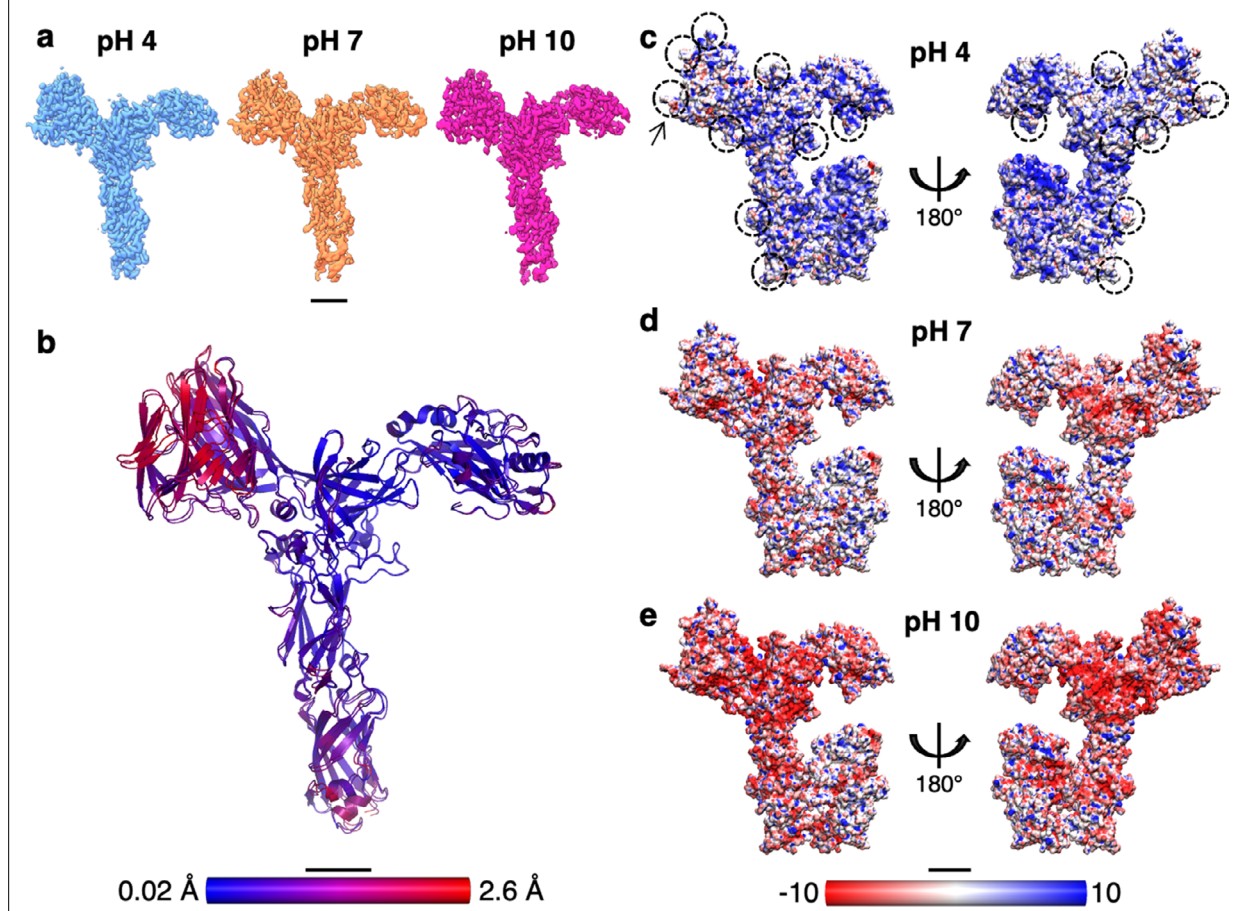

**Figure 3.** Structural comparison and electrostatic surface potentials of *S.acidocaldarius* SlaA at different pH conditions. (**a**) SlaA$_{30–1069}$ cryo electron microscopy (cryoEM) maps at pH 4 (light blue, res. 3.1 Å), pH 7 (orange, res. 3.9 Å), and pH 10 (magenta, res. 3.2 Å). (**b**) r.m.s.d. (root-mean-square deviation) alignment between SlaA$_{30–1069}$ atomic models at pH 4 and 10. Smaller deviations are shown in blue and larger deviations in red, with mean r.m.s.d. = 0.79 Å. Electrostatic surface potentials of SlaA at pH 4 (**c**), pH 7 (**d**), and pH 10 (**e**). Models include Alphafold-predicted C-terminal domains (in closed conformation). Surfaces are coloured in red and blue for negatively and positively charged residues, respectively. White areas represent neutral residues. In (**c**), some areas occupied by glycans are circled; the arrow points at one of the 6-sulfoquinovose residues displaying a negative charge at pH 4. Scale bar, 20 Å.

The online version of this article includes the following figure supplement(s) for figure 3:

**Figure supplement 1.** Relion processing workflow for pH 7 dataset.

**Figure supplement 2.** Relion processing workflow for pH 10 dataset.

**Figure supplement 3.** Impact of glycosylation on the electrostatic surface charge of SlaA at different pH values.

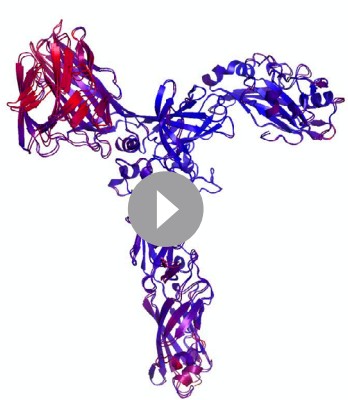

**Video 3.** Comparison of SlaA$_{30-1069}$ structure at pH 4 and 10. Root-mean-square deviation (r.m.s.d.) alignment between SlaA$_{30-1069}$ atomic models at pH 4 and 10. Smaller deviations are shown in blue and larger deviations in red, with mean r.m.s.d. = 0.79 Å, as in *Figure 3b*.

https://elifesciences.org/articles/84617/figures#video3

of 3.9 Å for SlaA at pH 7 and 3.2 Å for SlaA at pH 10 (*Figure 3a*; *Figure 1—figure supplement 3*). As for SlaA at pH 4, domains D5 and D6 were too flexible to be resolved in the cryoEM maps. Strikingly, the cryoEM maps of SlaA$_{30-1069}$ at the three pH conditions were virtually identical, demonstrating a remarkable pH stability of this protein. The mean r.m.s.d. (root-mean-square deviation) value of Cα atoms between the pH 4 and 10 structures was 0.79 Å (min. = 0.02 Å; max. = 2.6 Å) (*Figure 3b*; *Video 3*), confirming that SlaA$_{30-1069}$ maintains its structure unchanged across a surprisingly broad pH range. This suggests that a pH-induced conformational change or unfolding in SlaA$_{30-1069}$ is not the cause for S-layer disassembly. However, because D5 and D6 were not resolved in our map, a structural rearrangement affecting these domains remains a possibility.

A variation in pH can dramatically affect protein–protein interactions by changing the overall electrostatic surface potential of the protein complex (*Jensen, 2008*; *Zhang et al., 2011*). An analysis of the surface charges of SlaA, including the glycans, at pH 4, 7, and 10 revealed that the overall protein charge changes from positive at pH 4 to negative at pH 10 (*Figure 3c–e*). A comparison of the surface charge between glycosylated and non-glycosylated SlaA (*Figure 3—figure supplement 3*) showed that the glycans contribute considerably to the negative charge of the protein at higher pH values. This change in electrostatic surface potential may be a key factor in disrupting protein–protein interactions within the S-layer, causing its disassembly at alkaline pH.

## Atomic model of the *S. acidocaldarius* S-layer

In a previous study, we determined the location of SlaA and SlaB within the S-layer lattice by cryoET of whole cells and isolated S-layers (*Gambelli et al., 2019*). However, due to the limited resolution of the cryoEM maps and the lack of SlaA and SlaB atomic models, the details of the S-layer structure could not be explored. To address this knowledge gap, we performed cryoET and subtomogram averaging (STA) on *S. acidocaldarius* exosomes with improved imaging conditions and processing techniques. Exosomes are naturally secreted S-layer-encapsulated vesicles, with a diameter of about 90–230 nm (*Ellen et al., 2009*). To analyse the in situ structure of the S-layer, we performed STA using Warp (*Tegunov and Cramer, 2019*), Relion 3.1 (*Scheres, 2020*), and M (*Tegunov et al., 2021*) and obtained a cryoEM map at 11.2 Å resolution (*Figure 4—figure supplements 1 and 2*). We fitted our structure of SlaA into the S-layer map, which provided an atomic model of the assembled lattice (*Figure 4a, b*; *Figure 4—figure supplement 1d–i*).

When observed in the direction parallel to the membrane plane, the exosome-encapsulating S-layer displays a positive curvature, with an average curvature radius of ~ 84 nm (*Figure 4*). SlaA assembles into a sheet with a thickness of 95 Å. The long axes of the SlaA subunits are inclined by an angle of about ~28° with respect to the curved S-layer surface (*Figure 4d*). As a result of this inclination, effectively two zones in the SlaA assembly can be distinguished: an outer zone consisting of D1, D2, D3, and D4, and an inner zone formed by D5 and D6 (*Figure 4c, d*).

Six SlaA monomers assemble around a hexagonal pore of 48 Å in diameter (glycans not included) (*Figure 4a*). The D1 domains of these six monomers project into and define the shape of the hexagonal pore, together with the domains D3 and D4. The triangular pores that surround the hexagonal pores have a diameter of ~85 Å and are defined by the D2, D4, D5, and D6 domains of three SlaA molecules (*Figure 4e*). The D3 domain of each monomer overlaps with the D4 domain of the following monomer along the hexagonal ring in a clockwise fashion. The D5 and D6 domains of each SlaA

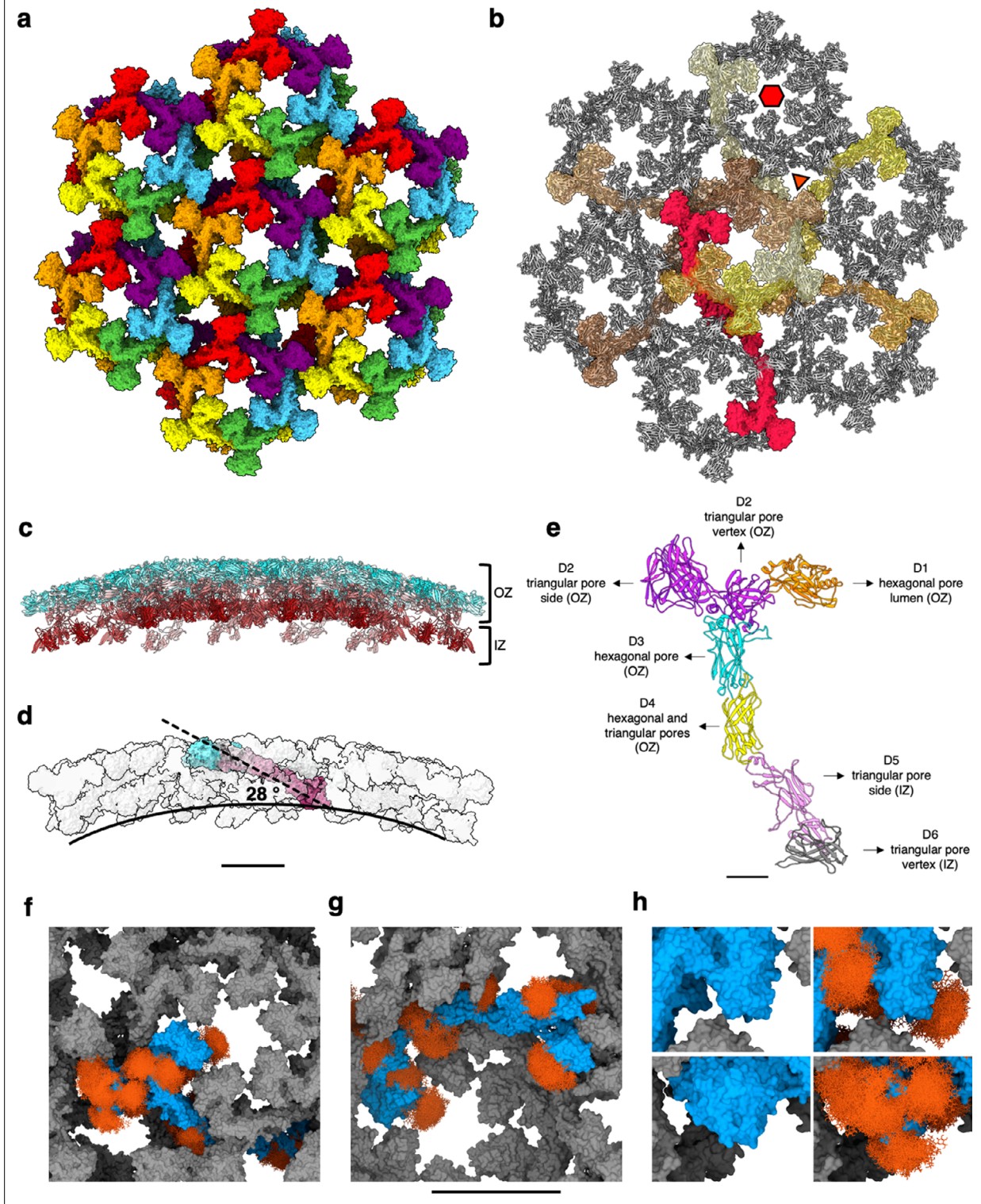

**Figure 4.** *S. acidocaldarius* SlaA assembly into exosome-bound S-layers. (**a**) Extracellular view of assembled SlaA monomers in rainbow colours and surface representation. (**b**) Extracellular view of assembled SlaA in ribbon representation with SlaA dimers forming a hexagonal pore highlighted in shades of red and yellow. Each dimer spans two adjacent hexagonal pores. (**c**) Side view of the SlaA lattice (blue, N-terminus; red, C-terminus). It is possible to distinguish an outer zone (OZ) formed by domain D1, D2, D3, and D4, and an inner zone (IZ) formed by domains D5 and D6. (**d**) One SlaA monomer (surface representation, N-terminus cyan, grey, C-terminus maroon) is highlighted within the assembled array. The long axis of each SlaA monomer (dashed line) is inclined by a 28° relative to the curved surface of the array (solid line). (**e**) The location of each SlaA domain within the S-layer. (**f–h**) SlaA glycans modelled with GlycoSHIELD in the assembled S-layer. (**f**) Shows the extracellular view; (**g**) shows the intracellular view; (**h**) shows insets

*Figure 4 continued on next page*

*Figure 4 continued*

of (**f**) at higher magnification without (left) and with (right) glycans. Glycans fill gaps unoccupied by the protein and significantly protrude into the lumen of the triangular and hexagonal pores. Scale bars in (**a–d, f–h**) 10 nm; in (**e**) 20 Å.

The online version of this article includes the following figure supplement(s) for figure 4:

**Figure supplement 1.** Subtomogram averaging of the S-layer on exosomes and fitting of SlaA.

**Figure supplement 2.** CryoET processing, STA and resolution estimation.

**Figure supplement 3.** Comparison between current and previously reported (*Gambelli et al., 2019*) *S. acidocaldarius* SlaA assembly models.

**Figure supplement 4.** Isolated SlaA-only S-layer from *S.acidocaldarius*.

subunit project towards the cell membrane. Two SlaA monomers dimerise through the D6 domains, with each SlaA dimer spanning two adjacent hexagonal pores (*Figure 4b, d, e*, *Figure 4—figure supplements 3 and 4*). Thus, protein–protein interactions between two adjacent hexagonal pores occur through the dimerising D6 domains of each SlaA dimer and the D2 domains of overlapping SlaA monomers. The SlaA dimer includes an angle of 160° between the two monomers, and has a total length of 420 Å (*Figure 4—figure supplement 3*). While SlaA was not resolved as a dimer in our SPA, we could confirm these dimers in tomograms of negatively stained S-layers (*Figure 4—figure supplement 4*), which show similar dimensions and structure as in our assembly model. Their co-existence with assembled S-layers may indicate that SlaA dimers are an intermediate of S-layer assembly or disassembly.

Modelling of glycan shields in the assembled structure showed that glycans fill large gaps seen between SlaA's globular domains and significantly protrude into the lumen of the triangular and hexagonal pores (*Figure 4f–h*). In the assembled S-layer, the interaction sites between SlaA largely occur via unglycosylated surfaces, leaving most glycans unaffected (*Figure 2—figure supplement 1*). Reduction of glycan conformational freedom is overall small between isolated and assembled SlaA monomers. Instead, the glycoshields appear to delineate protein–protein interfaces, which may 'guide' the self-assembly of the S-layer, substantiated by the fact that any restriction of glycan flexibility would be entropically unfavourable. Similarly, a glycan-guided assembly mechanism has been suggested for the assembly of cadherins in the desmosome (*Sikora et al., 2020*).

To get a handle on the structure of the entire S-layer, we used Alphafold v2.2.0 (*Jumper et al., 2021*) and SymmDock (*Schneidman-Duhovny et al., 2005*) and predicted the monomeric and trimeric SlaB structure. The predicted structure for one SlaB monomer consists of three N-terminal β-sandwich domains and a 132 amino acid long C-terminal α-helix (*Figure 5—figure supplement 1a*). As shown by our STA map (*Figure 5, figure supplement 3c, d*), SlaB forms a trimer. Alphafold v2.2.0 (*Jumper et al., 2021*) suggests that three SlaB molecules form a trimeric coiled-coil via their C-terminal α-helices, and their N-terminal β-domains fanning out into a propeller-like structure (*Figure 5a, b*; *Figure 5-figure supplement 1b*). This domain architecture agrees with the sequence-based molecular modelling described previously (*Veith et al., 2009*). The TMHMM-2.0 server predicted the C-terminal amino acids 448–470 as transmembrane helix. The hydrophobicity plot (*Figure 5—figure supplement 2e*) confirms a hydrophobic region corresponding to the predicted transmembrane helix (*Figure 5—figure supplement 2a, e*). The protein is predicted to have 14 *N*-glycosylation sites, of which six are located along the C-terminal α-helix (*Figure 5—figure supplement 2b–d*). The electrostatic surface potential calculated at pH 4 shows that the C-terminal α-helix is mostly neutral (*Figure 5—figure supplement 2f*). In contrast, the three β-sandwich domains have greater electrostatic potential. While D2 is mostly positive, D3 carries distinct negatively charged patches (*Figure 5—figure supplement 2f*). These patches may play a role in electrostatic interactions between SlaB's D3 domain and the mainly positively charged SlaA.

By combining SPA and STA with structural predictions, we built a complete *S. acidocaldarius* S-layer model (*Figure 5c–e*; *Figure 5—figure supplement 3*, *Video 4*) . The Alphafold predictions of the SlaB trimer superimposed remarkably well into the corresponding densities visible in our STA map at low threshold values, and flexible fitting using Namdinator (*Kidmose et al., 2019*) further improved the fit (*Figure 5—figure supplement 3*).

In the assembled lattice, SlaB trimers occupy alternating triangular pores around each hexagonal pore (*Gambelli et al., 2019*). The SlaB trimer has a tripod-like structure, with its long axis perpendicular to the planes formed by the membrane and SlaA. Three Ig-like domains branch away from the trimer's

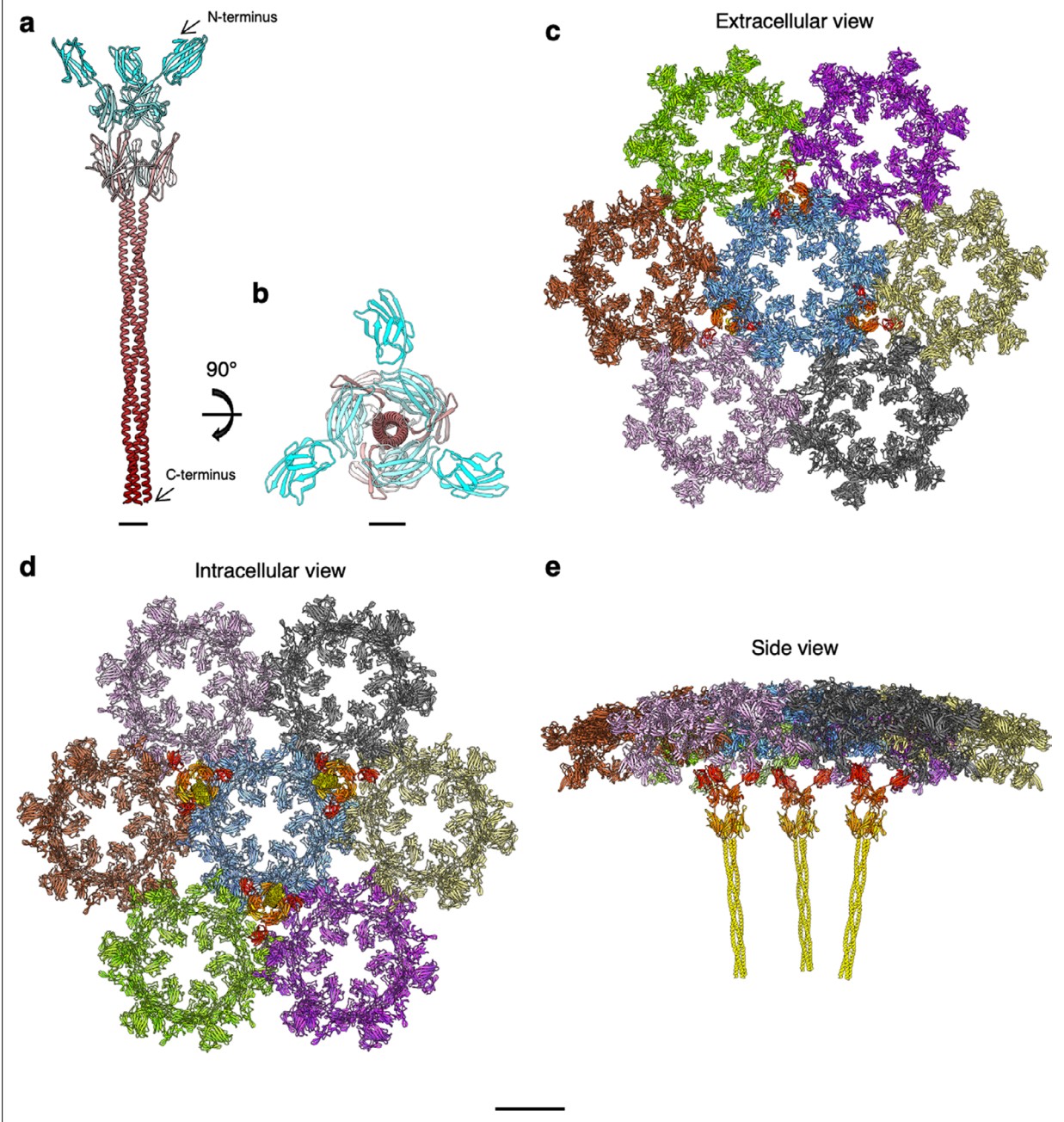

**Figure 5.** *S. acidocaldarius* S-layer assembly. (**a, b**) SlaB trimer (ribbon representation, N-terminus, cyan; C-terminus, maroon) as predicted by Alphafold v2.2.0 (*Jumper et al., 2021*). (**c–e**) Ribbon representation of the assembled SlaA and SlaB components of the S-layer. (**c**), (**d**), and (**e**) show the external face, the pseudo-periplasmic face, and a side view, respectively. SlaA proteins around each hexagonal pore are shown in different colours. SlaB trimers are shown in shades of yellow and orange (N-termini are red-orange shades and C-termini are yellow). Scale bar, (**a, b**) 20 Å; (**c–e**) 10 nm.

The online version of this article includes the following figure supplement(s) for figure 5:

**Figure supplement 1.** Alphafold v2.2.0 predictions of SlaB monomer and trimer.

**Figure supplement 2.** Structural prediction of *S. acidocaldarius* SlaB.

**Figure supplement 3.** Subtomogram average of the exosome-bound S-layer and SlaB fitting.

**Figure supplement 4.** Structure of archaeal and bacterial S-layer proteins.

**Figure supplement 5.** Stability and charge heatmaps for *S. acidocaldarius* SlaA$_{30-1069}$, SlaA, and SlaB.

**Figure supplement 6.** Stability and charge heatmaps for *C.crescentus* and *H. volcanii* S-layer proteins.

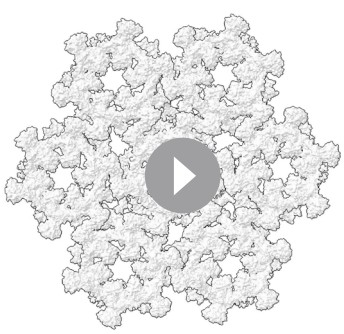

**Video 4.** Model of the assembled *S. acidocaldarius* S-layer.

https://elifesciences.org/articles/84617/figures#video4

symmetry axis and face the SlaA canopy, whereas three α-helices form a coiled coil, which at the predicted transmembrane region insert into the resolved exosome membrane (*Figure 5—figure supplement 3c*).

The lattice is a ~35-nm-thick macromolecular assembly, in which each SlaB trimer interacts with three SlaA dimers. This interaction may be mediated by the positively charged D6 dimerising domains of SlaA and the negatively charged N-terminal Ig-like D3 domains of SlaB.

## Discussion

The Sulfolobales S-layer lattice stands out from others because it is a two-component lattice, consisting of the S-layer-forming SlaA and the membrane anchor SlaB. In 2019, we reported on a first 3D map of the *S. acidocaldarius* S-layer obtained from STA on whole cells and isolated S-layer sheets (*Gambelli et al., 2019*). With the new information provided in the current study, we were able to improve on the model we proposed previously. The new data confirm the overall p3 S-layer lattice symmetry, in which the unit cell contains one SlaB trimer and three SlaA dimers (SlaB$_3$/3SlaA$_2$). Each SlaB trimer occupies alternating triangular pores and each SlaA dimer spans two adjacent hexagonal pores. Because each SlaB monomer interacts with the dimerisation domains of SlaA dimers, the SlaB trimer occupancy of all triangular pores would likely be unfavourable due to steric hindrance. Additionally, alternating SlaB throughout the array would reduce the protein synthesis costs for this protein by 50%. SlaB trimers occupying every second triangular pore also effectively create an S-layer with a variety of pore sizes, modulating the exchange of molecules with the environment.

Using exosomes and a new image processing approach, we were able to improve the resolution and eliminate the missing wedge in our subtomogram average of the *S. acidocaldarius* S-layer. The new map enabled us to build a revised model of the *S. acidocaldarius* S-layer assembly (*Figures 4 and 5*, *Video 4*). Here, the SlaA dimer (*Figure 4—figure supplement 3a*) spans an angle of 160° and extends over 42 nm, instead of 23 nm, as previously proposed (*Gambelli et al., 2019*). The increased length is largely a result of the unexpected positioning of domains D5 and D6, which were previously not accounted for (*Figure 4—figure supplement 3*).

SLPs of extremophilic archaea generally show a high degree of glycosylation, potentially aiding their survival in extreme environments (*Jarrell et al., 2014*). SlaA is predicted to contain 31 *N*-glycosylation sites (*Peyfoon et al., 2010*) and the SlaA$_{30-1069}$ cryoEM map showed 19 clear densities corresponding to *N*-glycosylation sequons. The cryoEM map contained densities for the complete hexasaccharide (*Peyfoon et al., 2010*; *Zähringer et al., 2000*) on the SlaA surface, as well as various glycan intermediates. We cannot rule out the possibility that our cryoEM map could not resolve the complete hexasaccharide on all sequons due to the flexibility of the glycans. Nevertheless, the presence of a heterogeneous family of glycans has previously been reported (*Peyfoon et al., 2010*), with nano-LC–ES-MS/MS used to analyse the structure of the glycans linked to the C-terminal portion of SlaA (residues 961–1395), and a heterogenous degree of glycosylation was observed including all intermediates from monosaccharide to complete hexasaccharide. The presence of a heterogeneous family of glycans has also been shown, for example, in the SLP of *H. volcanii* (*Abu-Qarn et al., 2007*) and the archaellum of *Methanothermococcus thermolithotrophicus* (*Kelly et al., 2020*). In archaea, the final step in protein glycosylation is catalysed by the oligosaccharyl transferase AglB (*Meyer and Albers, 2014*). The enzyme is promiscuous, meaning that AglB can load glycans of variable length on the lipid carrier (*Cohen-Rosenzweig et al., 2014*). While AglB is essential for the viability of *S. acidocaldarius* (*Meyer and Albers, 2014*), it remains to be determined whether the heterogenous composition of its glycans is to be attributed to AglB loading glycan precursors onto SlaA and/or glycan hydrolysis due to the harsh environmental conditions. A future study involving the genetic or

enzymatic ablation of glycosylation sites would shed more light on the roles that surface glycans play in S-layer structure, stability, and function.

Metal ions are often bound to SLPs and have recently been demonstrated to play a crucial role in S-layer assembly and cell-surface binding (*Cohen et al., 1991*; *Herdman et al., 2022*; *Baranova et al., 2012*; *Bharat et al., 2017*; *von Kügelgen et al., 2020*; *Lupas et al., 1994*; *Herrmann et al., 2020*). In the bacterium *C. crescentus*, whose S-layer has been investigated in detail, $Ca^{2+}$ ions are essential for intra- and inter-molecular stability of the S-layer lattice (*Herdman et al., 2022*; *Bharat et al., 2017*). Moreover, analogous results have been obtained for the S-layer of *Geobacillus stearothermophilus* (*Baranova et al., 2012*). The SLP of the archaeon *H. volcanii* has also been recently confirmed to bind cations (*von Kügelgen et al., 2021*). The *S. acidocaldarius* S-layer is no exception and its assembly is a $Ca^{2+}$-dependent process (*Gambelli et al., 2019*). Interestingly, the $SlaA_{30-1069}$ cryoEM map did not reveal any anomalous densities that could be attributed to ions. It is therefore possible that cations are harboured in the D5 and D6 domains that were not resolved, and/or at the protein–protein interfaces within the assembled lattice, which at this point cannot be defined at the side-chain level due to the limited resolution of our subtomogram average.

In a recent work, von Kügelgen et al. presented the structure of the *H. volcanii* S-layer (*von Kügelgen et al., 2021*). Therefore, the *H. volcanii* and *S. acidocaldarius* S-layers are currently the only two archaeal S-layers for which complete atomic models are available. *H. volcanii* is a halophilic archaeon of the Euryarchaeota phylum. As the *S. acidocaldarius* S-layer, the *H. volcanii* lattice also exhibits a hexagonal symmetry, but different architecture. The *H. volcanii* S-layer is constituted by a single glycosylated SLP named csg. SlaA (1424 residues) and csg (827 residues) both consist of six domains (*Figure 5—figure supplement 4b*). However, while all csg domains adopt Ig-like folds, SlaA is built up from domains of more complex topology. In csg, the domains are arranged linearly, whereas SlaA adopts an extended Y-shape (*Figure 5—figure supplement 4a, b*). Ig-like domains are widespread among SLPs in different archaeal phyla, including the order Sulfolobales (*von Kügelgen et al., 2021*). In fact, the SlaA protein of *Metallosphaera sedula* is predicted to consist of seven Ig-like domains (*Figure 5—figure supplement 4d*; *von Kügelgen et al., 2021*). The different domain architecture that we observe for *S. acidocaldarius* SlaA highlights the great divergence of S-layers among microorganisms.

Assembled csg forms hexagonal (13 Å), pentameric (6 Å), and trimeric (10 Å) pores much smaller than the hexagonal (48 Å) and trimeric (85 Å) pores of the *S. acidocaldarius* lattice. In both cases, the pore size is further reduced by glycans projecting into the pores. The glycans could regulate the permeability of the S-layer in a fashion similar to the hydrogel regulating the permeability of the nuclear pore complexes (*D'Angelo and Hetzer, 2008*). It is currently unknown which evolutionary parameters resulted in species-specific S-layer pore sizes. It may be speculated that, for example, these pores have co-evolved with and adapted their size according to certain secreted protein filaments, such as pili. *S. acidocaldarius* produces four such filaments – archaella (*Szabó et al., 2007*), A-pili (*Henche et al., 2012*), and UV-inducible pili and threads (*Fröls et al., 2008*). Of these four filaments, only threads, with a diameter of ~40 Å, would be able to pass through the hexagonal pores of the S-layer without the need for a widening of the pores or a partial S-layer disassembly. It is thus tantalising to speculate that the hexagonal S-layer pores have evolved to accommodate threads, perhaps as a scaffold for their assembly.

S-layers are intrinsically flexible structures as to encapsulate the cell entirely. In the case of *H. volcanii*, csg assembles around hexameric as well as pentameric pores on the surface of both exosomes and whole cells (*von Kügelgen et al., 2021*). Such pentameric 'defects' confer enough flexibility to the array to encase the cell in areas of low and high membrane curvature. Interestingly, we did not observe an analogous phenomenon for the *S. acidocaldarius* S-layer on whole cells or exosomes. However, symmetry breaks have been observed on S-layers isolated from whole cells at the edges where the lattice changes orientation (*Pum et al., 1991*). Furthermore, additional flexibility may be provided by the SlaA dimeric interface, as well as by loop regions linking the SlaA domains. In fact, only single loops link D1–D2, D3–D4, D4–D5, and D5–D6. While the reciprocal position of D3–D4 is stabilised by the disulphide bond ($Cys_{677}$–$Cys_{1017}$), the loops connecting D1–D2, D4–D5, and D5–D6 may allow the flexibility necessary for SlaA to be incorporated in this highly interwoven, yet malleable protein network.

Electrostatic interactions are critical for proper protein folding and function. Moreover, changes in surface charge have been shown to affect protein–protein interactions. Particularly, the pH plays a key role in determining the surface charge of proteins due to polar amino acid residues on the protein surface (*Jensen, 2008*; *Zhang et al., 2011*) . Remarkably, SlaA$_{30-1069}$ proved stable over a vast pH range and its tertiary structure remains virtually unchanged (*Figure 3*). Thus, we propose that is likely not pH-induced unfolding or conformational changes in SlaA that cause S-layer disassembly at alkaline pH.

The surface net charge of SlaA shifts from positive to negative when the pH is elevated from 4 to 10 (*Figure 3*, *Figure 3—figure supplement 3*).

The observed reversal in electrostatic potential at rising pH values is a manifestation of deprotonation of amino acid residues, as the concentration of hydrogen ions (H$^+$) in the solution decreases. The loss of protons can reduce or abolish the ability of side chains to form hydrogen bonds, and as a result, hydrogen bonds involving these groups can be weakened or broken. The weakening or abolishment of these bonds (in particular those involving acidic amino acids) could therefore be a key factor in pH-induced disassembly. Conversely, the lowering of the pH will re-protonate these residues, facilitate the formation of hydrogen bonds, and thus the assembly of the S-layer. However, it is important to note that the effects of pH on hydrogen bonding in proteins can be complex. Thus, further experimentation would be required to test this hypothesis.

Considerations regarding the pH stability of SlaA$_{30-1069}$ can be extended to the entirety of the protein using pH stability predictions, which suggest virtually no difference in pH-dependent protein stability across ionic strength and pH values for both SlaA$_{30-1069}$ and the full length SlaA protein (*Figure 5—figure supplement 5a–d*). This suggests that domains D5 and D6 equally do not unfold at alkaline pH. Analogous predictions of protein stability were obtained for SlaB (*Figure 5—figure supplement 5e, f*), where the net charge is slightly positive across pH 2–8. For comparison, we ran the same predictions on the *C. crescentus* and *H. volcanii* S-layer proteins RsaA and csg, respectively (*Figure 5—figure supplement 6*). Among SlaA, SlaB, RsaA, and csg, we observe that SlaA and SlaB are expected to be the most stable at different pH values. Notably, csg is most stable at acidic pH and progressively less so at neutral and alkaline pH. This prediction is confirmed by experimental data (*Rodrigues-Oliveira et al., 2019*), which additionally showed pH-dependent protein folding rearrangements and protein unfolding. It is to be considered that this prediction does not include glycosylation (*Hebditch and Warwicker, 2019*), which enhances S-layer stability, especially in the case of Sulfolobales (*Jarrell et al., 2014*; *Meyer and Albers, 2014*; *Meyer et al., 2011*; *Yurist-Doutsch et al., 2008*). The resilience of SlaA at temperature and pH shifts can likely be attributed to two main factors: the high glycosylation level, and the fact that ~56% of SlaA$_{30-1069}$ has a defined secondary structure, which allows the formation of intramolecular bonds (*Vogt et al., 1997*).

S-layers are often necessary for the survival of microorganisms in nature but can also be of great interest for synthetic biology. Therefore, a greater understanding of their structural details will strongly aid their nanotechnological uses, which have already shown remarkable potential in biomedical (*Lanzoni-Mangutchi et al., 2022*; *Luo et al., 2019*; *Fioravanti et al., 2022*) and environmental applications (*Charrier et al., 2019*; *Pallares et al., 2022*; *Zhang et al., 2021*; *Schuster and Sleytr, 2021*).

## Methods

### *S. acidocaldarius* strains and growth conditions

Cells of *S. acidocaldarius* strain MW001 were grown in basal Brock medium* at pH 3 (*Brock et al., 1972*) as previously described (*Gambelli et al., 2019*). Briefly, cells were grown at 75°C, 150 rpm, until an OD600 of >0.6 was reached. Cells were then centrifuged at 5000 × *g* (Sorvall ST 8R) for 30 min at 4°C. The cell fraction was stored at −20°C for S-layer isolation, whereas the supernatant was stored at 4°C for exosomes isolation.

*Brock media contain (per l): 1.3 g (NH$_4$)2SO$_4$, 0.28 g KH$_2$PO$_4$, 0.25 g MgSO$_4$·7H$_2$O, 0.07 g CaCl$_2$·2H$_2$O, 0.02 g FeCl$_2$·4H$_2$O, 1.8 mg MnCl$_2$·4H$_2$O, 4.5 mg Na$_2$B$_4$O$_7$·10H$_2$O, 0.22 mg ZnSO$_4$·7H$_2$O, 0.05 mg CuCl$_2$·2H$_2$O, 0.03 mg NaMoO$_4$·2H$_2$O, 0.03 mg VOSO$_4$·2H$_2$O, 0.01 mg CoSO$_4$·7H$_2$O, and 0.01 mg uracil.

## S-layer isolation and disassembly

The S-layer isolation and disassembly were performed as previously described (*Gambelli et al., 2019*). Briefly, frozen cell pellets from a 50 ml culture were incubated at 40 rpm (Stuart SB3) for 45 min at 37°C in 40 ml of buffer A (10 mM NaCl, 1 mM phenylmethylsulfonyl fluoride, 0.5% sodium lauroylsarcosine), with 10 μg/ml DNase I. The samples were pelleted by centrifugation at 18,000 × *g* (Sorvall Legend XTR) for 30 min and resuspended in 1.5 ml of buffer A, before further incubation at 37°C for 30 min. After centrifugation at 14,000 rpm for 30 min (Sorvall ST 8R), the pellet was purified by resuspension and incubation in 1.5 ml of buffer B (10 mM NaCl, 0.5 mM MgSO$_4$, 0.5% sodium dodecyl sulfate [SDS]) and incubated for 15 min at 37°C. To remove SlaB from the assembled S-layers, washing with buffer B was repeated three more times. Purified Sla-only S-layers were washed once with distilled water and stored at 4°C. The removal of SlaB was confirmed by SDS/polyacrylamide gel electrophoresis (PAGE) analysis. S-layers were disassembled by increasing the pH to 10 with the addition of 20 mM NaCO$_3$ and 10 mM CaCl$_2$ and incubated for 2 hr at 60°C at 600 rpm (Thermomixer F1.5, Eppendorf).

## SlaA purification

After disassembly, the sample containing SlaA was further purified using gel filtration chromatography. A total of 100 μl containing 10 mg/ml of disassembled protein were loaded onto a Superdex 75 Increase 10/300 GL (GE Healthcare) using 300 mM NaCl for elution. At the end of the run, the fractions containing SlaA were dialysed against 30 mM acetate buffer (0.1 M CHCOOH, 0.1 M CH$_3$COONa) at pH 4, 150 mM Tris–HCl at pH 7, or 20 mM NaCO$_3$ at pH 10, with the aim to compare the SlaA protein structure at different pH values. The purity of the fractions was assessed by SDS/PAGE analysis and negative staining with 1% uranyl acetate on 300 mesh Quantifoil copper grids with continuous carbon film (EM Resolutions).

## CryoEM workflow for SPA

### Grid preparation

The purified SlaA samples at pH 4 and 10 (3 μl of ~0.1 mg/ml) were applied to 300 mesh copper grids with graphene oxide-coated lacey carbon (EM Resolutions) without glow discharge. Grids were frozen in liquid ethane using a Mark IV Vitrobot (Thermo Fisher Scientific, 4°C, 100% relative humidity, blot force 6, blot time 1 s) with Whatman 597 filter paper. The purified SlaA at pH 7 was applied to glow discharged R 1.2/1.3 300 mesh copper grids with holey carbon. The freezing procedure was kept the same as for the samples at pH 4 and 10 besides the blot time of 2 s.

### Data collection

Micrographs were collected on a 200 kV FEI Talos Arctica TEM, equipped with a Gatan K2 Summit direct detector using EPU software (Thermo Fisher Scientific) (*Supplementary file 1a*). Data were collected in super-resolution at a nominal magnification of ×130,000 with a virtual pixel size of 0.525 Å at a total dose of ~60 e⁻/Å (*Fagan and Fairweather, 2014*). A total of 3687 movies (44 fractions each), 3163 movies (44 fractions each), and 5046 movies (60 fractions each), with a defocus range comprised between −0.8 and −2.4 μm, were collected for samples at pH 4, 7, and 10, respectively.

### Image processing

Initial steps of motion correction (MotionCor 2; *Li et al., 2013*) and Contrast Transfer Function (CTF) estimation (CTF-find 4; *Rohou and Grigorieff, 2015*) were performed in Relion 3.0 (*Zivanov et al., 2018*) and Relion 3.1 (*Scheres, 2020*) for datasets at pH 4 and 7, whereas Warp (*Tegunov and Cramer, 2019*) was used for the pH 10 dataset. Further steps of 2D and 3D classification, refinement, CTF refinement, and polishing were performed using Relion 3.1. For a detailed workflow of the three datasets see *Figure 1—figure supplement 1*, *Figure 3—figure supplements 1 and 2*. The refined maps were post-processed in Relion 3.1 as well as using DeepEMhancer (*Sanchez-Garcia et al., 2021*). The produced maps had a resolution of 3.1, 3.9, and 3.2 Å at pH 4, 7, and 10, respectively, by gold-standard FSC 0.143.

## Model building and validation

The SlaA atomic model was built de novo using the cryoEM map at pH 4 in Buccaneer (*Cowtan, 2006*), refined using REFMAC5 (*Murshudov et al., 2011*) and rebuilt in COOT (*Emsley et al., 2010*). The glycans were modelled in COOT with the refinement dictionary for the unusual sugar 6-sulfoquinovose prepared using JLigand (*Lebedev et al., 2012*). This atomic model was then positioned into the cryoEM maps at pH 10 and 7 using ChimeraX (*Pettersen et al., 2021*) and refined using REFMAC5 and COOT. All models were further refined using Isolde (*Croll, 2018*) and validated using Molprobity (*Chen et al., 2010*) in CCP4 (*Winn et al., 2011*).

### Exosome isolation

*S. acidocaldarius* exosomes were isolated from the supernatant obtained after cell growth. The procedure was adapted from *Ellen et al., 2009*. The supernatant was split into 8 fractions and exosomes were pelleted in two runs of ultracentrifugation (Optima LE-80K, Beckman Coulter) at $125,000 \times g$ for 45 min at 4°C. The pellet was resuspended in 2 ml (per fraction) of the supernatant and ultracentrifuged (Optima MAX-TL, Beckman Coulter) at 12,000 rpm (TLA55 rotor, Beckman Coulter) for 10 min at 4°C. The pellet (containing intact cells and cell debris) was discarded, and the supernatant was ultracentrifuged (Optima MAX-TL, Beckman Coulter) at 42,000 rpm (TLA55 rotor, Beckman Coulter) for 90 min at 4°C. The pellet containing the isolated exosomes was resuspended in MilliQ water at a concentration of 15 mg/ml. The purity of the sample was assessed by negative staining with 1% uranyl acetate on 300 mesh Quantifoil copper grids with continuous carbon film (EM Resolutions).

### CryoEM workflow for STA

#### Grid preparation

The isolated exosomes were mixed 1:1 with 10 nm colloidal gold conjugated protein A (BosterBio) and 3 μl droplets were applied four times on glow discharged 300 mesh Quantifoil copper R2/2 grids (EM Resolutions). The grids were blotted with 597 Whatman filter paper for 4 s, using blot force 1, in 95% relative humidity, at 21°C, and plunge-frozen in liquid ethane using a Mark IV Vitrobot (FEI).

#### Data collection

Micrographs were collected on two microscopes: a 200 kV FEI Talos Arctica TEM, equipped with a Gatan K2 Summit direct detector and a 300 kV Thermo Fisher Titan Krios G3 with a Thermo Fisher Falcon 4i direct detector and SelectrisX energy filter, both using the Tomo 4 package. Tilt series on the Talos/K2 were collected in super-resolution at a nominal magnification of ×63,000 with a virtual pixel size of 1.105 Å at a total dose of ~83 e⁻/Å². The tilts were collected from −20° to 60° in 3 degree steps (2 fractions per tilt). Tilt series on the Krios/Falcon 4 were collected as conventional MRC files at 4k × 4k, nominal magnification of ×64,000 and a pixel size of 1.9 A at a total dose of ~83 e⁻/Å². Tilts were collected from −60° to 60° in 3 degree steps in a dose-symmetric scheme with groupings of 2 (6 fractions per tilt). A nominal defocus range between −4 and −6 μm was used for both collections. A total of 86 positions were collected, 28 on the Talos and 58 on the Krios.

#### Electron cryo-tomography and STA

Initial STA was performed using only data collected on the Talos. Motion correction was performed using the IMOD (*Kremer et al., 1996*) program alignframes. IMOD was also used for the tomogram reconstruction. Initial particle picking on all 28 tomograms was performed using seedSpikes and spikeInit as part of the PEET software package (*Nicastro et al., 2006*) with a total of 12,010 particles picked. For initial STA, the picked particles were CTF corrected and extracted using the Relion STA pipeline (*Bharat and Scheres, 2016*). 2D classification, initial model generation, 3D classification and initial refinements were all performed using Relion 3.1 (*Scheres, 2020*). A resolution of 16.1 Å was reached using 1313 particles and C3 symmetry.

For higher-resolution averaging, the tilt series from both datasets were processed using the Warp–Relion–M pipeline (*Tegunov et al., 2021*). Motion correction and CTF estimation of the movies were performed in Warp (*Tegunov and Cramer, 2019*). The poor quality tilts were excluded and Aretomo (*Zheng et al., 2022*) was used to provide alignments on the resulting tilt series stacks for tomogram reconstruction in Warp. Deconvolved tomograms were used to visualise the exosomes and, as above,

seedSpikes and spikeInit were used to generate initial particle coordinates for the S-layer. A total of 22,950 particles were picked and subsequently extracted in Warp at a pixel size of 10 Å/px. The two datasets were processed separately with several rounds of refinement and classification until they reached a resolution of 20 Å with C3 symmetry. For both datasets, the 16.1 Å map from the initial averaging was used, low-pass filtered to 60 Å. The two maps were visually compared and found to be different sizes, so the pixel size of the Talos data was adjusted. The tomograms were reprocessed and particles re-extracted at 10 Å/px then refined until a resolution of 20 Å was again achieved. The particles were combined together then refined in M to a resolution of 16 Å (C3 symmetry). The particles were extracted at a pixel size of 5 Å/px. Further refinement and 3D classification resulted in a 14 Å resolution. A final iteration in M resulted in a resolution of 11.2 Å with 2771 particles used in the refinement.

The model of the assembled S-layer was built by initial rigid body fitting the SlaA structure determined by SPA into the subtomogram average using ChimeraX (*Pettersen et al., 2021*). The C-terminal domains of SlaA that were predicted in Alphafold2 (*Jumper et al., 2021*) were then added to each SlaA. Hereby, only SlaA in the extended conformation could be reconciled with the map. Next, the SlaB trimers were predicted in Alphafold2 and fitted into the trimeric stalks that connected the S-layer canopy with the membrane. Finally, the model was refined using Namdinator (*Kidmose et al., 2019*), a molecular dynamics-based flexible fitting software.

## Structure analysis and presentation

The electrostatic potential of the protein was derived using APBS (Adaptive Poisson-Boltzmann Solver) (*Jurrus et al., 2018*) based on the PARSE force field for the protein as available through PDB2PQR (*Dolinsky et al., 2007*). Where available, the charges of the glycans were assigned based on the GLYCAM force field (*Kirschner et al., 2008*); charges of the hydrogens were combined with their central heavy atom. The charge assignment depends on the bonding topology, that is occupied linkage positions. *Supplementary file 1b* summarises the mapping of residue from the structure file to GLYCAM residue names. For residue styrene maleic acid or anhydride (SMA), charge assignments are not available from the GLYCAM force field; these were derived based on restrained electrostatic potential (RESP) calculations conducted for the methoxy derivatives on the HF/6-1G*//HF/6-31G* level of theory and employing a hyperbolic restraint equal to 0.010 in the charge fitting step (*Breneman and Wiberg, 1990*; *Dupradeau et al., 2010*). The total charge of the newly derived residue was constrained to $-0.8060$ e and $-1$ e for the 1-substituted and 1,4-substituted SMA (referred to as SG0 and SG4 in *Supplementary file 1c, d*), respectively, in agreement with the conventions of the GLYCAM force field. In assembling the final charge assignment, the charge of the linking ND2 atom of the glycosylated Asn residues of the protein was altered to compensate for the polarisation charge of the attached saccharide unit. The electrostatic charge was visualised using VMD (*Humphrey et al., 1996*) (http://www.ks.uiuc.edu/Research/vmd/).

The structure of *S. acidocaldarius* SlaA was visualised using UCSF Chimera (*Pettersen et al., 2004*), Chimera X v.1.3 and v1.4 (*Pettersen et al., 2021*), and Pymol (*Delano, 2002*). The structural domains of SlaA were assigned using SWORD (*Postic et al., 2017*).

Heatmaps for net charge, and pH and ionic strength-dependent protein stability were obtained using Protein-Sol (https://protein-sol.manchester.ac.uk/) (*Hebditch and Warwicker, 2019*). For SlaB the signal peptide was predicted using InterPro (*Blum et al., 2021*), the transmembrane region was predicted using TMHMM-2.0 (*Krogh et al., 2001*), the *N*-glycosylation sites (sequons N-X-S/T) were predicted using GlycoPP v1.0 (*Chauhan et al., 2012*).

## Molecular dynamics simulations

Conformation arrays of glycans were grafted on protein structure using GlycoSHIELD (*Gecht et al., 2021*). In brief, glycan systems (GlcNAc[2],Man[2],QuiS[1],Glc[1] N-linked to neutralised glyc–Asp–gly tripeptides) were modelled in CHARMM-GUI (*Jo et al., 2008*) and solvated using TIP3P water models in the presence of 150 mM NaCl and configured for simulations with CHARMM36m force fields (*Park et al., 2019*; *Huang et al., 2017*). Molecular dynamics simulations were performed with GROMACS 2020.2 and 2020.4-cuda (*Abraham et al., 2015*) in mixed GPU/CPU environments. Potential energy was first minimised (steepest descent algorithm, 5000 steps) and were equilibrated in the canonical ensemble. 1 fs time steps and Nose–Hoover thermostat were used. Atom positions and dihedral

angles were restrained during the equilibration, with initial force constants of 400, 40, and 4 kJ/mol/$nm^2$ for restraints on backbone positions, side-chain positions, and dihedral angles, respectively. The force constants were gradually reduced to 0. Systems were additionally equilibrated in NPT ensemble (Parrinello–Rahman pressure coupling with the time constant of 5 ps and compressibility of $4.5 \times 10^{-5}$ $bar^{-1}$) over the course of 10 ns with a time step of 2 fs. Hydrogen bonds were restrained using LINCS algorithm. During the production runs, a velocity-rescale thermostat was used and the temperature was kept at 351 K. Production runs were performed for a total duration of 3 µs and snapshots of atom positions stored at 100 ps intervals.

Glycan conformers were grafted using GlycoSHIELD with a distance of 3.25 Å between protein α-carbons and glycan ring-oxygens. Glycan conformers were shuffled and subsampled for representation of plausible conformations on displayed renders. Graphics were generated with ChimeraX (*Pettersen et al., 2021*).

## Acknowledgements

We acknowledge Ufuk Borucu for help with data collection, and the GW4 Facility for High-Resolution Electron Cryo-Microscopy, funded by the Wellcome Trust (202904/Z/16/Z and 206181/Z/17/Z) and BBSRC (BB/R000484/1). We also acknowledge Alexander Neuhaus for assistance with data analysis. We thank IDRIS for the allocation of high-performance computing resources (allocations #2020-AP010711998 and #2021-A0100712343 to CH). For this project, LG, BD, MM, MG, RC, and KS were funded by the European Research Council (ERC) under the European Union's Horizon 2020 research and innovation programme (grant agreement No. 803894). MM was also funded by a BBSRC New Investigator Research Grant (BB/R008639/1) to VG and RC by the University of Exeter and a Wellcome Trust Seed Award in Science (210363/Z/18/Z) awarded to VG and RC, as well as a Wellcome Trust Seed Award in Science (212439/Z/18/Z) awarded to BD and RC. MS was funded under Dioscuri, a programme initiated by the Max Planck Society, jointly managed with the National Science Centre in Poland, and mutually funded by Polish Ministry of Education and Science and German Federal Ministry of Education and Research (UMO-2021/03/H/NZ1/00003)

## Additional information

### Funding

| Funder | Grant reference number | Author |
| --- | --- | --- |
| European Research Council | 803894 | Lavinia Gambelli<br>Mathew McLaren<br>Rebecca Conners<br>Kelly Sanders<br>Matthew C Gaines<br>Bertram Daum |
| Wellcome Trust | 10.35802/210363 | Rebecca Conners<br>Vicki AM Gold |
| Wellcome Trust | 10.35802/212439 | Bertram Daum<br>Rebecca Conners |
| Agence Nationale de la Recherche | ANR-16-CE16-0009-01 | Cyril Hanus |
| Agence Nationale de la Recherche | ANR-21-CE16-0021-01 | Cyril Hanus |
| Leverhulme Trust | RPG-2020-261 | Daniel Kattnig |
| Biotechnology and Biological Sciences Research Council | BB/R008639/1 | Rebecca Conners<br>Mathew McLaren<br>Vicki AM Gold |
| Polish Ministry of Education and Science | UMO-2021/03/H/NZ1/00003 | Mateusz Sikora |

| Funder | Grant reference number | Author |
|---|---|---|
| German Federal Ministry of Education and Research | UMO-2021/03/H/NZ1/00003 | Mateusz Sikora |

The funders had no role in study design, data collection, and interpretation, or the decision to submit the work for publication. For the purpose of Open Access, the authors have applied a CC BY public copyright license to any Author Accepted Manuscript version arising from this submission.

## Author contributions

Lavinia Gambelli, Data curation, Formal analysis, Validation, Investigation, Visualization, Writing – original draft; Mathew McLaren, Data curation, Formal analysis, Validation, Investigation, Visualization, Methodology; Rebecca Conners, Cyril Hanus, Formal analysis, Investigation, Methodology; Kelly Sanders, Investigation, Methodology; Matthew C Gaines, Formal analysis; Lewis Clark, Formal analysis, Investigation; Vicki AM Gold, Resources, Writing – original draft; Daniel Kattnig, Formal analysis, Methodology; Mateusz Sikora, Methodology; Michail N Isupov, Formal analysis, Validation, Investigation, Methodology; Bertram Daum, Conceptualization, Resources, Data curation, Formal analysis, Supervision, Funding acquisition, Validation, Investigation, Visualization, Writing – original draft, Project administration

## Author ORCIDs

Lavinia Gambelli https://orcid.org/0000-0002-2257-6364
Mathew McLaren http://orcid.org/0000-0002-6636-7409
Lewis Clark http://orcid.org/0009-0003-9917-3912
Vicki AM Gold http://orcid.org/0000-0002-6908-0745
Daniel Kattnig http://orcid.org/0000-0003-4236-2627
Mateusz Sikora https://orcid.org/0000-0003-1691-4045
Michail N Isupov http://orcid.org/0000-0001-6842-4289
Bertram Daum http://orcid.org/0000-0002-3767-264X

## Decision letter and Author response

Decision letter https://doi.org/10.7554/eLife.84617.sa1
Author response https://doi.org/10.7554/eLife.84617.sa2

# Additional files

## Supplementary files

• Supplementary file 1. CryoEM statistics. (**a**) Statistics of data collection, 3D reconstruction, and validation. (**b**) Mapping of glycan residues from the structure file to residues of the GLYCAM force field or the newly charge-derived SG0 and SG4 residues, representing the 1-substituted and 1,4-substituted SMA. (**c**) RESP charges derived for residue SG0 on the HF/6-31G*//HF/6-31G* level of theory (see Methods for details). (**d**) RESP charges derived for residue SG4 on the HF/6-31G*//HF/6-31G* level of theory (see Methods for details).

• MDAR checklist

## Data availability

The atomic coordinates of SlaA were deposited in the Protein Data Bank (https://www.rcsb.org/) with accession numbers PDB-7ZCX, PDB-8AN3, and PDB-8AN2 for pH 4, 7 and 10, respectively. The cryoEM maps were deposited in the EM DataResource (https://www.emdataresource.org/) with accession numbers EMD-14635, EMD-15531 and EMD-15530 for pH 4, 7 and 10, respectively. The sub-tomogram averaging map of the S-layer has been deposited in the EMDB (EMD-18127) and models of the hexameric and trimeric pores in the Protein Data Bank under accession codes PDB-8QP0 and PDB-8QOX, respectively. Other structural data used in this study are: H. volcanii csg (PDB ID: 7PTR, https://doi.org/10.2210/pdb7ptr/pdb), and C. crescentus RsaA (N-terminus PDB ID: 6T72, https://doi.org/10.2210/pdb6t72/pdb, C-terminus PDB ID: 5N8P, https://doi.org/10.2210/pdb5n8p/pdb). The raw image data used in this study have been deposited to the Electron Microscopy Public Image Archive (EMPIAR) under accession numbers EMPIAR-11788 (single particle datasets at pH 4, 7 and 10) and EMPIAR-11888 (tomography datasets).

The following datasets were generated:

| Author(s) | Year | Dataset title | Dataset URL | Database and Identifier |
|---|---|---|---|---|
| Gambelli L, Isupov MN, Daum B | 2023 | S-layer protein SlaA from Sulfolobus acidocaldarius at pH 4.0 | https://www.emdataresource.org/EMD-14635 | EMDataResource, EMD-14635 |
| Gambelli L, Isupov MN, Daum B | 2023 | S-layer protein SlaA from Sulfolobus acidocaldarius at pH 7.0 | https://www.emdataresource.org/EMD-15531 | EMDataResource, EMD-15531 |
| Gambelli L, Isupov MN, Daum B | 2023 | S-layer protein SlaA from Sulfolobus acidocaldarius at pH 4.0 | https://www.rcsb.org/structure/7ZCX | RCSB Protein Data Bank, 7ZCX |
| Gambelli L, Isupov MN, Daum B | 2023 | S-layer protein SlaA from Sulfolobus acidocaldarius at pH 7.0 | https://www.rcsb.org/structure/8AN3 | RCSB Protein Data Bank, 8AN3 |
| Gambelli L, McLaren MJ, Daum B | 2024 | S-layer of archaeon Sulfolobus acidocaldarius by subtomogram averaging | https://www.emdataresource.org/EMD-18127 | EMDataResource, EMD-18127 |
| Gambelli L, Isupov MN, Daum B | 2023 | S-layer protein SlaA from Sulfolobus acidocaldarius at pH 10.0 | https://www.rcsb.org/structure/8AN2 | RCSB Protein Data Bank, 8AN2 |
| Gambelli L, Isupov MN, Daum B | 2023 | S-layer protein SlaA from Sulfolobus acidocaldarius at pH 10.0 | https://www.emdataresource.org/EMD-15530 | EMDataResource, EMD-15530 |
| Gambelli L, McLaren M, Isupov M, Conners R, Daum B | 2024 | Two-component assembly of SlaA and SlaB S-layer proteins of Sulfolobus acidocaldarius | https://www.rcsb.org/structure/8QOX | RCSB Protein Data Bank, 8QOX |
| Gambelli L, McLaren M, Isupov M, Conners R, Daum B | 2024 | A hexamer pore in the S-layer of Sulfolobus acidocaldarius formed by SlaA protein | https://www.rcsb.org/structure/8QP0 | RCSB Protein Data Bank, 8QP0 |
| Gambelli L, McLaren MJ, Sanders K, Gaines M, Clark L, Gold VAM, Kattnig D, Sikora M, Hanus C, Isupov M, Daum B | 2024 | Sulfolobus acidocaldarius s-layer SlaA | https://www.ebi.ac.uk/empiar/EMPIAR-11788 | EMPIAR, EMPIAR-11788 |
| Gambelli L, McLaren MJ, Sanders K, Gaines M, Clark L, Gold VAM, Kattnig D, Sikora M, Hanus C, Isupov M, Daum B | 2024 | Sulfolobus acidocaldarius s-layer SlaA cryoET dataset | https://www.ebi.ac.uk/empiar/EMPIAR-11888 | EMPIAR, EMPIAR-11888 |

The following previously published datasets were used:

| Author(s) | Year | Dataset title | Dataset URL | Database and Identifier |
|---|---|---|---|---|
| von Kuegelgen A, Bharat TAM | 2021 | Structure of hexameric S-layer protein from Haloferax volcanii archaea | https://doi.org/10.2210/pdb7ptr/pdb | Worldwide Protein Data Bank, 10.2210/pdb7ptr/pdb |
| von Kuegelgen A, Bharat TAM | 2020 | Structure of the RsaA N-terminal domain bound to LPS | https://doi.org/10.2210/pdb6t72/pdb | Worldwide Protein Data Bank, 10.2210/pdb6t72/pdb |
| Bharat TAM, Kureisaite-Ciziene D, Lowe J | 2017 | S-layer protein RsaA from C. crescentus | https://doi.org/10.2210/pdb5n8p/pdb | Worldwide Protein Data Bank, 10.2210/pdb5n8p/pdb |

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
