## [Editor Report]

The manuscript brings important new insights in S-layer structure and assembly, providing a first experimental model of a crenarchaeotal S-layer. The work provides solid evidence for the S-layer architecture and its role in supporting the archaeal cell envelope. This will be of broad interest to microbiologists and biotechnologists seeking to understand the biological role and technological application of these enigmatic membrane support structures.

---

## [Decision Letter]

**Decision letter after peer review:**

Thank you for submitting your article "CryoEM reveals the structure of an archaeal two-component S-layer" for consideration by *eLife*. Your article has been reviewed by 2 peer reviewers, and the evaluation has been overseen by Han Remaut as Reviewing Editor and Reviewer #1, and Volker Dötsch as the Senior Editor.

Essential revisions:

1) Provide additional validation and description of the S-layer model derived from the 3D cryoET.

2) Deposit the cryoET maps and S-layer model in an appropriate repository.

3) Review the clarity, labelling, and message of the Figures.

4) Review the balance of citations on S-layer (structural) biology, including bacterial S-layers.

*Reviewer #2 (Recommendations for the authors):*

1) Please add scale bars in Supplementary figure 2 d-f and Supplementary figure 15 f.

2) Line 70: Please correct: p2 forms oblique lattice symmetry and not, as indicated the square one.

3) A lot of references are incomplete (Refs. 8, 24, 26, 30, 32, 34, 49, 56, 64, 83, 91).

4) The title is somehow misleading as it indicates that solely cryoEM was sufficient to reveal the structure of this archaeal two-component S-layer.

[Editors’ note: further revisions were suggested prior to acceptance, as described below.]

Thank you for resubmitting your work entitled "Structure of the two-component S-layer of the archaeon Sulfolobus acidocaldarius" for further consideration by *eLife*. Your revised article has been evaluated by Volker Dötsch (Senior Editor) and a Reviewing Editor.

The manuscript has been substantially improved but there are some remaining editorial issues that need to be addressed, as outlined below:

In your final manuscript, please take into account the request to provide a more balanced review of bacterial S-layers.

*Reviewer #2 (Recommendations for the authors):*

The revised manuscript has been improved according to the reviewer's comments in terms of indicating the different angles between the SlaA longitudinal axis and the membrane and the between two SlaA monomers forming a dimer; the assessment of the assembly state by negative stain electron microscopy; the brief statement to the discussion to clarify SlaA dimerization; the add scale bars in Supplementary figures; the p2 oblique lattice symmetry; the updated references; and the simplified title of the manuscript.

However, there are still many references incomplete (Refs. 71, 73, 75, 77, 78).

Unfortunately, in the Introduction Section reporting on prokaryotes (line 42 to 72), no incorporation of more references on bacterial S-layers are distinguishable. References 1 and 6 do not report on S-layer at all (In Ref. 1 it is clearly stated: "S-layers and capsules, …, are beyond the scope of this review."). References 2, 3, 4, 5, 8, 9 are exclusively dealing with archaeal S-layers. In reference 9, the symmetry of S-layers for selected archaea is reported, but there is no explanation given what one understands under this specific term, not to mention that there are no schematic drawings in this reference. Thus, the task of an essential revision of the balance of citations on S-layer (structural) biology, including bacterial S-layers has not been done.

---

## [Author Response]

Essential revisions:1) Provide additional validation and description of the S-layer model derived from the 3D cryoET.

Done.

2) Deposit the cryoET maps and S-layer model in an appropriate repository.

Done.

3) Review the clarity, labelling, and message of the Figures.

Done.

4) Review the balance of citations on S-layer (structural) biology, including bacterial S-layers.

Done.

Reviewer #2 (Recommendations for the authors):1) Please add scale bars in Supplementary figure 2 d-f and Supplementary figure 15 f.

We have added the missing scale bars in Figure 2. There is no supplementary figure 15 f, so we assume that the reviewer meant 15 c and d and have added scale bars there (see new Figure 4—figure supplement 4).

2) Line 70: Please correct: p2 forms oblique lattice symmetry and not, as indicated the square one.

We thank the reviewer for spotting this error and have corrected it.

3) A lot of references are incomplete (Refs. 8, 24, 26, 30, 32, 34, 49, 56, 64, 83, 91).

We thank the reviewer for notifying us about this and updated the references.

4) The title is somehow misleading as it indicates that solely cryoEM was sufficient to reveal the structure of this archaeal two-component S-layer.

We have simplified the title to omit any ambiguity: Structure of the two-component S-layer of the archaeon *Sulfolobus acidocaldarius.*

[Editors’ note: what follows is the authors’ response to the second round of review.]

The manuscript has been substantially improved but there are some remaining editorial issues that need to be addressed, as outlined below:In your final manuscript, please take into account the request to provide a more balanced review of bacterial S-layers.Reviewer #2 (Recommendations for the authors):The revised manuscript has been improved according to the reviewer's comments in terms of indicating the different angles between the SlaA longitudinal axis and the membrane and the between two SlaA monomers forming a dimer; the assessment of the assembly state by negative stain electron microscopy; the brief statement to the discussion to clarify SlaA dimerization; the addition of scale bars in Supplementary figures; the p2 oblique lattice symmetry; the updated references; and the simplified title of the manuscript.

We gratefully acknowledge the reviewer’s appreciation for the improvements that we made based on their suggestions.

However, there are still many references incomplete (Refs. 71, 73, 75, 77, 78).

Many thanks for spotting this – we now fixed those references.

Unfortunately, in the Introduction Section reporting on prokaryotes (line 42 to 72), no incorporation of more references on bacterial S-layers are distinguishable. References 1 and 6 do not report on S-layer at all (In Ref. 1 it is clearly stated: "S-layers and capsules, …, are beyond the scope of this review.").

We replaced reference #1 for doi.org/10.1016/j.tim.2020.09.009, which provides an overview of on bacterial and archaeal cell envelops and S-layer protein structures.

To clarify, reference #6 supports our statement that S-layers are thought to mediate virus-entry into bacteria and archaea in some cases. This reference states:

“It is likely that archaeal viruses, like their bacterial counterparts, recognize and bind cell wall components, such as S-layer proteins and sugar moieties.”

We therefore prefer to keep this reference in our manuscript.

However, we also incorporated a second reference (https://doi.org/10.1128/mbio.01833-22), which demonstrates that the S-layer of *Haloferax gibbonsii* acts as primary receptor for the virus HFTV1.

References 2, 3, 4, 5, 8, 9 are exclusively dealing with archaeal S-layers.

Acknowledged. We have now incorporated additional references discussing bacterial S-layers: https://doi.org/10.1038/nrmicro3213 and 10.1111/1574-6976.12063.

In reference 9, the symmetry of S-layers for selected archaea is reported, but there is no explanation given what one understands under this specific term, not to mention that there are no schematic drawings in this reference.

We replaced reference 9 with a more general overview of S-layers that also contains schematic drawings explaining S-layer symmetry (doi: 10.1111/1574-6976.12063).

We trust that updating these references now generates a more balanced overview of bacterial and archaeal S-layers in the introduction of our paper.